# Combination Therapies with PRRT

**DOI:** 10.3390/ph14101005

**Published:** 2021-09-30

**Authors:** Anna Yordanova, Hojjat Ahmadzadehfar

**Affiliations:** Department of Nuclear Medicine, Klinikum Westfalen, 44309 Dortmund, Germany; hojjat.ahmadzadehfar@ruhr-uni-bochum.de

**Keywords:** PRRT, NET, combination therapies, personalized medicine, somatostatin analogues, chemotherapy, molecular targeted treatment, liver radioembolization, Lutetium-177, Yttrium-90

## Abstract

Peptide receptor radionuclide therapy (PRRT) is a successful targeted radionuclide therapy in neuroendocrine tumors (NETs). However, complete responses remain elusive. Combined treatments anticipate synergistic effects and thus better responses by combining ionizing radiation with other anti-tumor treatments. Furthermore, multimodal therapies often have a balanced toxicity profile. To date, few studies have evaluated the effect of combination therapies with PRRT, some of them phase I/II trials. This review will focus on several clinically tested, tailored approaches to improving the effects of PRRT. The aim is to help clinicians in the treatment planning of NETs to choose the most effective and safe treatment for each patient in the sense of personalized medicine. Current promising combination partners of PRRT are somatostatin analogues (SSAs), chemotherapy, molecular targeted treatment, liver radioembolization, and dual radionuclide PRRT (Lutetium-177-PRRT combined with Yttrium-90-PRRT).

## 1. Introduction

Neuroendocrine tumors (NETs) are heterogeneous neoplasia that are often diagnosed in the metastasized stage (range 40 to 76%), making them challenging to manage [1,2,3]. Guideline-oriented treatment options normally target only one specific pathway of the cell cycle. Such options are not always suitable for heterogeneous clones and can eventually result in treatment resistance [4,5,6].

Peptide receptor radionuclide therapy (PRRT) is proven to be an effective and safe treatment (EMA 2017 and FDA 2018 approved) that significantly prolongs survival and improves quality of life. However, according to prospective phase III study data, it has a limited response rate of only 18% [7,8,9]. Furthermore, since PPRT is not a curative treatment, patients eventually relapse. If recurrent tumors still have adequate somatostatin receptor (SSTR) expression, there is a good chance that salvage PRRT [10,11] will be beneficial. On the contrary, dedifferentiated NETs with loss of SSTR expression have a poor outcome with short survival following monotherapy [12]. Thus, combined treatment is a promising option for targeting heterogeneous tumors and avoiding accumulated toxicity. However, the data on combined treatment is still limited.

This review summarizes current data from clinically proven combination treatments with PRRT and aims to help physicians choose a tailored treatment approach for patients with NETs. Combination partners with possible synergistic therapeutic effects seem to be dual-PRRT radiolabeling, liver radioembolization, “non-radiolabeled” somatostatin analogues (SSAs), chemotherapy (e.g., capecitabine/temozolomide), molecular targeted treatment (e.g., everolimus), [^131^I]I-metaiodobenzylguanidine (MIBG), and external beam radiotherapy (EBRT).

### Combination Treatment Decision Making

As represented in Figure 1, before starting a treatment, physicians should not only prioritize maximizing tumor response and patient survival but also minimize adverse events and patient morbidity. Substantial factors to consider in this decision-making are the age and health condition of the patient; genetic factors, tumor characteristics such as the origin, localization, size, and immunohistochemical proliferation marker Ki67; and tumor uptake in molecular imaging such as SSTR-positron emission tomography (PET) and [^18^F]F-Fluorodeoxyglucose (FDG). Furthermore, an institution’s access to multidisciplinary medical care and medical center experience are important features for treatment planning [13,14,15]. Figure 2 represents the various antitumor effects of combination partners of PRRT. The objective response rates, PFS, OS and adverse events of combination partners of PRRT are listed in Table 1.

Dual-imaging SSTR-PET and FDG-PET can help clinicians plan individualized treatments [16]. Chan et al. developed a scoring system (NETPET grade) to distinguish between tumors with both SSTR- and FDG-positive lesions, only SSTR- or FDG-positive lesions, and both SSTR- and FDG-negative lesions. The NETPET grade is prognostic for survival and can help to determine which patients are likely to benefit from combination therapy, such as PRRT and chemotherapy [17,18].

## 2. Dual PRRT

Several agents are used for PRRT in advanced somatostatin receptor positive NETs. The essential components of radiopharmaceuticals are an SSA, which targets the somatostatin receptors, a radioisotope, and a linking molecule (chelator) between them. The pioneer in PRRT was diethylenetriaminepentaacetic acid-(DTPA)-D-[Phe1]- Octreotide labelled with Indium-111. Indium-111 emits Auger electrons and conversion electrons with a potentially cytotoxic effect after internalization of the radiolabeled agent. Furthermore, the gamma emission of Indium-111 enables imaging of SSTR-positive tumors. However, standard treatment with [^111^In]In-DTPA-Octreotide rarely resulted in objective responses (<10%) [53]. High activity treatment seems to be more effective (13% complete remission, 20% partial remission), but the time to disease progression remains relatively short, with a median of 9 months [54]. Nowadays, the most commonly used and studied agents for the therapy are [[^90^Y]Y-DOTA,D-Phe1,Tyr3]-octreotide ([^90^Y]Y-DOTATOC) and [[^177^Lu]Lu-DOTA0,Tyr3]-octreotate ([^177^Lu]Lu-DOTATATE).

PRRT with Lutetium-177 has less detrimental effects in large, bulky NET metastases with nonhomogeneous SSTR distribution in comparison to Yttrium-90 due to its lower energy and shorter penetration range (maximum 2–4 mm vs. 11 mm). On the one hand, the radiation energy of Yttrium-90 is not completely absorbed in smaller tumors. On the other, Lutetium-177 alone may fail to induce a complete remission in large tumors. Therefore, PRRT combined with Lutetium-177 and Yttrium-90 might be a solution in such cases [19,20].

According to several similar studies, tandem PRRT leads to better results than monotherapy with Yttrium-90-PRRT: overall survival (OS) of 5.51 years vs. 3.96 years with [^90^Y]Y-Octreotide alone (*p* = 0.006), a high response rate of 42%, and still comparable toxicity (2% MDS, 2% grade 3 nephrotoxicity, and 7% grade 3/4 hematotoxicity) [20,32,33,34,35,36,37]. However, there are no comparative studies between the dual PRRT and the FDA-approved treatment LUTATHERA^®^.

In a recent report from a Warsaw study group, patients reached an OS of 7.46 years, calculated from the first tandem PRRT, and 10.61 years from the first NET diagnosis. In the subgroup analysis, patients with G1 and large bowel NET had the longest PFS/OS. The risk of progression in the first 2 years was 42% [55].

A phase II comparative study is now recruiting NET patients who are receiving PRRT with Yttrium-90 (4 × 3.7 GBq), PRRT with Lutetium-177 (4 × 5.55 GBq) or mixed therapy (4 × 3.7 GBq). The treatment will consist of 4 cycles 8 ± 2 weeks apart. Approximately 150 participants with GEP-NETs and non-GEP-NETs, including bronchopulmonary NETs, pheochromocytoma/paragangliomas and NETs of an unknown primary, are expected to be included in the study. The analysis will strive to have a long follow-up (up to 8 years) to determine progression-free survival (PFS), OS, and safety (CI: NCT04029428).

Novel combined dual PRRT is the combination of alpha-emitter PRRT and beta-emitter PRRT. Tumor hypoxia is a significant factor for resistance of cancer cells to β-emitters. Thus, α-emitters can be advantageous in some cases due to a higher energy transfer and smaller penetration range. In the clinical setting alpha-emitter PRRT is applied in case of tumor resistance to conventional PRRT. Most studied isotopes for alpha radionuclide therapy are Bismuth-213 and Actinium-225. [56,57,58,59,60]. Newer promising developments are SSTR-agonists labelled with Lead-212, such as AlphaMedix^®^ [61,62,63]. The first results of the prospective phase I trial show good tolerability in PRRT-naïve patients (CI: NCT03466216).

## 3. Chemotherapy

Low-dose chemotherapy may have a radiosensitizing effect via increased DNA damage, inhibition of DNA repair, cell proliferation arrest, tumor cell re-oxygenation, synchronization of the cell cycle, or apoptosis [21]. The most commonly used radiosensitizing substances are capecitabine, temozolomide, and 5-fluorouracil (5-FU).

G3-neuroendocrine neoplasia (NEN) with Ki67 < 55% seems to be less responsive to chemotherapy than G3-NEN with Ki67 > 55% [64]. The OS of patients with Ki67 < 55% treated with chemotherapy in the NORDIC trial was 14 months (Sorbye et al., 2013). In contrast, patients with SSTR-positive tumors with Ki67 < 55% treated with combined PRRT and chemotherapy (PRCRT) reached, according to retrospective analyses, an OS of 46 months [64,65]. Other retrospective studies have reported a disease control rate of 55–70% in multiple relapsed and extensively pre-treated NETs [66,67].

The Melbourne study group analyzed 68 patients after combined PRRT with 5-FU. The first cycle PPRT was given alone. The 5-FU (200 mg/m^2^/d) started 4 days before the second PRRT and continued for 3 weeks. Objective responses in computed tomography (CT) were seen in 25% of cases, and an additional 7% of cases showed minor responses. The majority of patients had stable disease [44].

A study conducted in Rotterdam evaluated the safety of four cycles of PPRT (7.4 GBq [^177^Lu]Lu-Octreotate) combined with capecitabine (1650 mg/m^2^ per day for two weeks). Of the seven patients included in the study, there was one case of grade 3 anemia and one case of grade 3 thrombocytopenia. No other severe adverse events were observed [38].

An Australian study (phase II) investigated the efficacy of patients after PRRT combined with capecitabine under a similar protocol. About one-fourth of patients had an objective response, only 6% progressed, and the majority had stable disease [39]. In a similar study by Nicolini et al., the combined PRRT plus capecitabine in 37 selected patients reached both somatostatin receptor- and FDG-positive GEP-NETs (Ki67% < 55%), PFS of 31 months; OS after a median follow-up of 38 months was not reached. The most common G3/G4 toxicities were neutropenia (11%), fatigue (5%), and diarrhea (5%). According to RECIST 1.1, 30% of patients responded, and 55% were stable [40].

Better responses with similar toxicities have been observed with a combination of PRRT and CAPTEM: capecitabine (14 days of 1500 mg/m^2^) and temozolomide (5 days of 200 mg/m^2^. About 3% of patients had grade 3 nausea, and 6% had grade 3 neutropenia. About 53–70% had an objective response. The proportion of complete responses was relatively high, at 13–15% [41,42]. Patients achieved a median PFS of 48 months, and OS after a median follow-up of 33 months was not reached [42]. Rarely there can be life-threatening neutropenia. An interesting report from Berlin described a case of neutropenic sepsis accompanied by fungal pulmonary infection and necrotizing mastitis about four weeks after the first cycle of combined PRRT plus CAPTEM. Still, the treatment has been continued after stabilization of the patient and resulted in a nearly complete response [68]. Surprisingly, recent retrospectively generated data from Mumbai showed no significant difference in PFS after combined treatment of PRRT plus CAPTEM compared to CAPTEM alone in patients with both SSTR- and FDG-positive G2/G3-NETs. In the multivariate analysis, CAPTEM alone or with PRRT significantly improved (*p* = 0.04) the outcomes of dual positive NET patients with a Ki-67 index > 5% [43].

A multicenter randomized clinical trial from Australia is currently recruiting patients with G1/G2 NETs to compare the benefits of PRRT vs. CAPTEM vs. combined PRRT and CAPTEM. The combination treatment will start with capecitabine 750 mg/m^2^ on days 1–14, followed by 7.8 GBq PPRT with [^177^Lu]Lu-Octreotate on day 10 and, at the end of the cycle, temozolomide 75mg/m^2^ b.i.d. on days 10–14. The treatment will include 4 cycles, 8 weeks apart (CI: NCT02358356). Another ongoing prospective trial evaluating combined treatment should be completed soon (CI: NCT02736448).

## 4. Somatostatin Analogues

Treatment with SSAs can result in the upregulation of SSTR [22]. The overexpression of the tumor targets SSTR2 in NETs could increase the effectiveness of PRRT without increasing the toxicity profile.

SSAs, like PRRT, have a high affinity to SSTR2 and therefore might act competitively in binding the tumor cells of NETs or lead to saturation. In current protocols, long-acting SSA should be discontinued about 4 weeks before PRRT to avoid interactions with radiolabeled SSAs [69]. Several studies suggest that discontinuation of somatostatin agonists prior to PRRT/SSTR-PET/CT is not necessary. In fact, the uptake in the tumor was unaffected or slightly increased, and the uptake in normal tissues, such as spleen and liver, decreased [70,71,72]. The explanation for this effect might be the saturation of SSTRs in healthy tissues and the upregulation of SSTR in tumor cells [6]. However, more investigations to determine the interaction between PRRT and SSA are needed to change standard protocols.

The NETTER-1 phase III trial showed, in advanced midgut NETs, that PRRT combined with SSA significantly prolonged PFS compared to SSA alone [8]. Recent analysis showed clinically improved median OS of 48 vs. 36 months compared to the control arm. However, the difference was not statistically significant between both groups, probably because of the high rate of cross-over-treatment in the study (36%) [73]. A debatable point of the NETTER-1 study is whether the effect of PRRT has been potentiated by SSA. A recent retrospective study aimed to examine whether a superior survival benefit of PRRT combined with SSAs exists over monotherapy with PRRT. The analysis showed that SSA combined with PRRT and/or as a maintenance treatment after PRRT significantly prolongs survival compared to PRRT alone (PFS 48 months vs. 27 months; OS 91 months vs. 47 months). Furthermore, the death event rates in patients with combined treatment were lower: 26% vs. 63% [46]. A multicenter retrospective trial PRELUDE examined the effects of the SSA lanreotide autogel/depot (LAN) combined with PRRT in progressive NETs. No increased adverse drug reactions were reported. More than one-third of patients had an objective response, and 95% were, at the last follow-up visit (12 months post-treatment), still progression-free. Naturally, these are retrospective data and might be prone to bias. However, if a patient tolerates treatment with SSA, there is no reason to withdraw SSA before or after PRRT. Furthermore, SSA may improve the outcomes of patients who receive PRRT [46]. To validate these data, prospective studies in a larger population using standardized treatment protocols are warranted.

## 5. Fractionated External Beam Radiotherapy

In preclinical studies, radiation seems to upregulate SSTR2 and thus enhances the effect of PRRT [23]. Additionally, an abscopal effect with the triggering of immuno-mediated antitumor effects can occur [24,26]. A Würzburg study group tested, in 10 patients with unresectable meningiomas, the combination of one cycle of PRRT with subsequent fractionated EBRT. No relevant toxicity was observed. After a median follow-up of 105 months, seven patients were stable (PFS 108 months), and 3 patients progressed (PFS 26 months). As a possible method of enhancing the efficacy of the treatment, Hartrampf et al. suggested a “sandwich approach” with PRRT prior to and post-EBRT. This approach might enhance the SSTR expression of meningioma cells and boost the antitumor effect after the second PRRT. Another option could be an intraarterial instead of intravenous application of PRRT due to the high arterial perfusion of meningiomas [47]. This has been shown in a case report by Braat et al. that demonstrated an 11-fold increase in tumor uptake and an estimated absorbed dose of 51 Gy in meningiomas [74]. Still, more comparative studies with larger cohorts are needed to verify the benefits of combined EBRT and PRRT.

## 6. Liver Radioembolization

The most common site of metastatic spread of NETs is the liver (>80% of patients with metastatic disease) (Rimakki et al.). About 70% of patients with large lesions have extensive metastatic liver involvement, which is associated with poor prognosis [75]. A recent subanalysis of the NETTER-1-study showed that the PFS in NET patients with large tumor lesions (>3 cm in diameter) was significantly shorter (*p* = 0.022) than in patients with small lesions [76]. The reason for the worse effect of large lesions with Lutetium-177 (Lu-177) may be Lu-177′s maximum tissue penetration of only 2–4 mm. For a better outcome, PRRT with Yttrium-90 (Y-90) or radioembolization of liver metastases with Y-90 or Holmium-166 (Ho-166) might be helpful. In a comparative analysis, patients treated with radioembolization plus PRRT showed a better disease control rate (87% vs. 67%) and superior OS compared to the radioembolization alone group (68 months vs. 35 months) [50]. According to recent data, radioembolization in NETs after initial PRRT is feasible, with objective responses of 16% after Y-90 and 43% after Ho-166 radioembolization. Each study included one patient with fatal radioembolization-induced liver disease [48,49]. Therefore, more data on such combination therapies in larger cohorts, especially regarding hepatotoxicity, are needed. 

## 7. Everolimus

Everolimus is an oral agent that targets the mammalian target of rapamycin (mTOR) with growth-inhibitory and anti-angeogenic effects. As an anti-angeogenic drug, everolimus prevents tumors from building abnormal vessels, leading to a more sufficient blood supply and better delivery of anti-tumor drugs to the tumor [27,28]. Phase III studies RADIANT-3 and RADIANT-4 demonstrated that patients with gastroenteropancreatic NETs can be effectively and safely treated with everolimus [32,77].

Clinical experience with the combined treatment of everolimus and PRRT is rather limited. In a phase I study, patients received escalating doses of everolimus: 5 to 10 mg/d for 24 weeks, and PRRT each 8 weeks (4 cycles). The maximum tolerated dose of everolimus in combination with PRRT was 7.5 mg/d [51]. A preclinical comparison study in mice with SSTR-positive tumors showed better responses and longer survival after combined treatment with PRRT plus everolimus than both the placebo group and everolimus alone. However, there was no significant difference in the outcome between the combined treatment and PRRT alone [78]. Clinical data comparing the combined treatment of PRRT and everolimus with monotherapies is still lacking.

## 8. [131. I]I-MIBG

MIBG labeled with iodine-131/123 is a guanethidine analog of norepinephrine and a well-established theranostic agent for some NETs, including pheochromocytoma/paraganglioma and neuroblastoma [29].

A Phase 1 clinical trial investigated the treatment combination of [^90^Y]Y-DOTATOC plus [^131^I]I-MIBG in three patients with advanced, progressive NETs. Patients received [^90^Y]Y-DOTATOC on day 1 and [^131^I]I-MIBG on day 2, and the treatment was repeated after 10-12 weeks. The trial had several problems. First, the study protocol was no longer up-to-date, because the study approval had lasted 6 years and thus had to be adapted to account for new scientific knowledge. For example, the first planned amino acid solution was too emetogenic and had to be replaced with a lysine/arginine solution. Second, this experimental phase 1 trial had to compete with the FDA-approved and commercially available treatment LUTATHERA^®^. This negatively impacted the number of participating patients. In summary, the study pointed out that, in some cases, combined PRRT with [^131^I]I-MIBG might be advantageous by delivering higher tumor doses. This treatment achieved, in selected patients, a tumor dose increase of 43–83% compared to [^90^Y]Y-DOTATOC alone. According to these results, the combined treatment seems to be safe. There was one case of grade 3 thrombocytopenia, but no other dose-limiting toxicities were observed in this very small group of patients. In the depicted design of the trial, the combined treatment consisted of significant specific activity of [^131^I]I-MIBG (AZEDRA^®^, FDA approval June 2018) and [^177^Lu]Lu-DOTATATE [52].

## 9. Promising Future Combination Therapies

Sunitinib malate is an orally available small-molecule tyrosine kinase inhibitor that arrests tumor growth. This is achieved by targeting the vascular endothelial growth factor receptor (VEGFR), platelet-derived growth factor (PDGFR), and receptor tyrosine kinase KIT. In a prospective phase III trial, sunitinib showed, in patients with pancreatic NETs, a doubling of PFS compared to the placebo—12 months vs. 6 months. However, there was no significant difference in the median OS: 39 months for sunitinib vs. 29 months for placebo. It is possible that this was due to the study’s cross-over design [30].

A current randomized phase II study aims to compare the efficacy of sunitinib and PRRT in advanced metastatic pancreatic NETs. The estimated completion date of the study is October 2023 (CI: NCT02230176). It will be interesting to determine the results of the cross-over groups, as sunitinib seems to be a potential radiosensitizer and might improve the effects of PRRT [79]. However, to date, there are no substantial clinical data on the combined treatment of PRRT and sunitinib. 

An open label prospective phase I/II trial aimed to explore the combination of the anti-PD-1 checkpoint inhibitor nivolumab with PRRT in 9 patients with pretreated advanced SSTR-positive lung cancer, predominantly small cell lung cancer. Preliminary results (CI: NCT03325816) show that low-level activity PRRT (3,7 GBq LUTATHERA^®^) two-monthly and nivolumab two-weekly for a period of 6 months had no dose limiting toxicity. Combined treatment with higher level activity PRRT (7,4 GBq LUTATHERA^®^) led to a single case of grade 3 rash. Most common grade 3 adverse event was lymphopenia (3 patients). From the 6 patients with measurable disease one patient achieved a partial response and two patients achieved a stable disease. The trial status is currently completed [80]. Further results of response and survival data should follow.

Another promising combination partner of PRRT in the future might be small-molecule poly (ADP-ribose) polymerase-1 inhibitors (PARPi), such as talazoparib (Talzenna^®^). In preclinical studies, PARPi combined with PRRT increased DNA double-strand tumor breaks and increased survival compared to PRRT as a monotherapy. Further clinical evaluations of this combination treatment in NET patients are warranted [31,81,82].

## 10. Conclusions

Combination therapies in NETs, especially combinations with PRRT as a concept of personalized medicine, seem to have a high potential for guideline implementation and translation to clinical applications. The goal is to develop tailored therapeutic protocols combining effectiveness along with safety. Despite the existence of several FDA-approved treatments for NETs, such as PRRT with LUTATHERA^®^, SSAs, Everolimus, Sunitinib, and [^131^I]I-MIBG (AZEDRA^®^), combinations of different drugs in NETs are still insufficiently studied. The most solid clinical experience has been the combination of PRRT and SSA (NETTER-1, phase III study), which has proven to be more effective than SSA alone [8]. Furthermore, retrospective data show that combined PRRT and SSA is more effective than PRRT alone [46]. Retrospective and phase I/II trials have revealed positive results of dual-PRRT [20,32,33,34,35,36,37] or combined PRRT with chemotherapy, such as CAPTEM [41,42,43]. 

However, further preclinical studies and molecular examinations are needed to better understand the synergistic effects of substances involved in different pathways in the induction of proliferation arrest of NETs. Furthermore, large, prospective studies are needed to corroborate the effects of different combined treatments in patients. 

In conclusion, even though there are still limited preclinical and prospective data on combination treatments with PRRT, this approach will certainly play an increasing role in the management of metastatic NETs. Interdisciplinary treatment planning is important for choosing the best treatment option for each patient without compromising their quality of life.

## Figures and Tables

**Figure 1 pharmaceuticals-14-01005-f001:**
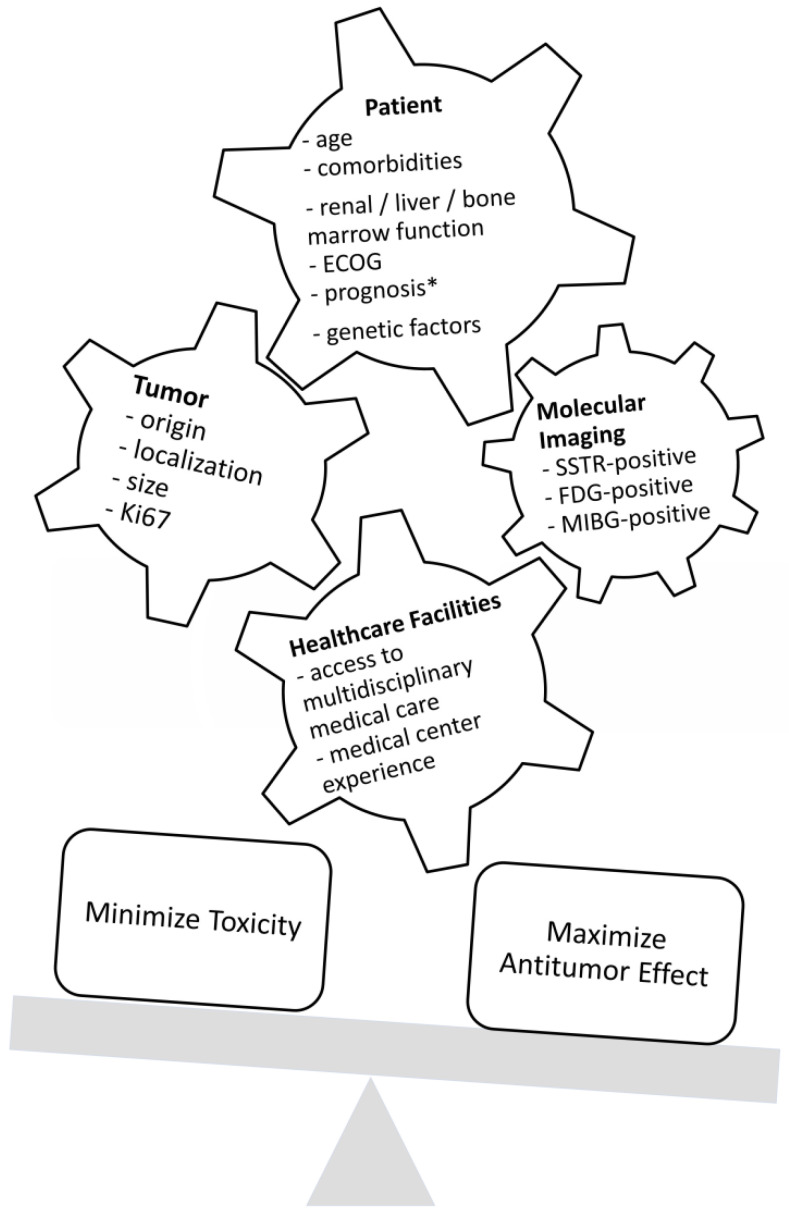
Factors that influence decision-making regarding treatment. ECOG = Eastern Cooperative Oncology Group Performance Status; SSTR = somatostatin receptor; FDG = fluorodeoxyglucose; MIBG = metaiodobenzylguanidine; * life expectancy of at least 3 months.

**Figure 2 pharmaceuticals-14-01005-f002:**
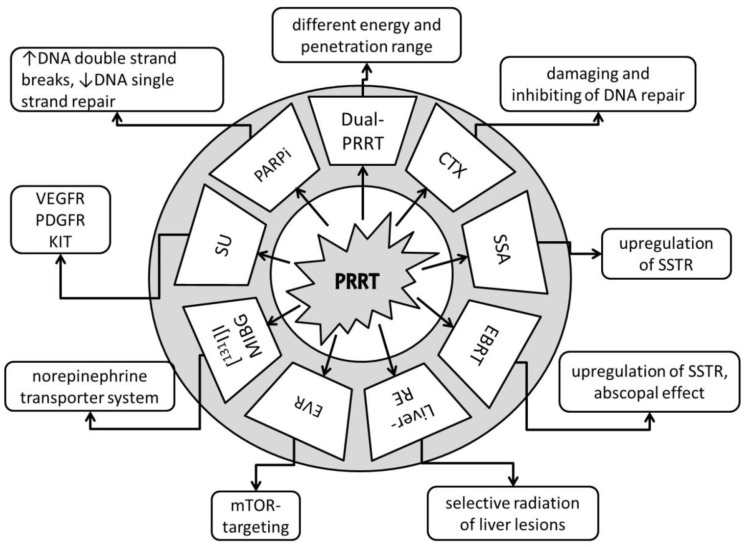
Anti-tumor effects of combination partners of PRRT. Dual-PRRT = dual-radionuclide peptide receptor therapy: a combination of different energy and penetration range levels to better target metastatic lesions with different sizes and nonhomogeneous somatostatin receptor (SSTR) distributions [19,20]. CTX = chemotherapy: damaging and inhibiting DNA repair, cell proliferation arrest, tumor cell reoxygenation, and synchronization of the cell cycle or apoptosis [21]. SSA = somatostatin receptor analogues: upregulation of SSTR, increasing number of targets for PRRT [22]. EBRT = fractionated external beam radiotherapy: upregulation of SSTR, increasing number of targets for PRRT, potential abscopal effect with triggering of immuno-mediated antitumor effects [23,24]. Liver-RE = liver radioembolization: selective radiation of liver tumor lesions; potential abscopal effect with triggering of immuno-mediated antitumor effects [25,26]. EVR = everolimus: targets the mammalian target of rapamycin (mTOR), with growth-inhibitory and anti-angeogenic effects [27,28]. [^131^I]I-MIBG = [^131^I]I-metaiodobenzylguanidine: targets the norepinephrine transporter system [29]. SU = sunitinib: tumor growth arrest via targeting of the vascular endothelial growth factor receptor (VEGFR), platelet-derived growth factor (PDGFR), and receptor tyrosine kinase KIT [30]. PARPi = poly (ADP-ribose) polymerase-1 inhibitors: increases DNA double-strand breaks; blocks DNA single-strand repair [31].

**Table 1 pharmaceuticals-14-01005-t001:** Efficacy and safety of combination treatment with PRRT.

Combination Partner	ORR (%)	OS (Month)	PFS (Month)	SAE (%)	Ref
Dual PRRT Lu-177 and Y-90	42	66–127	-	2% MDS, 2% nephrotoxicity, 7% hematotoxicity	[20,32,33,34,35,36,37]
Capecitabine	24–30	not reached	31	<15% anemia/thrombocytopenia/neutropenia 5% fatigue/diarrhea	[38,39,40]
CAPTEM	53–70	not reached	22–48	6% neutropenia, 3% nausea	[41,42,43]
5-fluorouracil	25	not reached	-	-	[44]
SSA	37	91	48	3% hepatotoxicity	[45,46]
EBRT	0	not reached	108	0%	[47]
Liver embolization	16 (Y-90) 43 (Ho-166)	42–68	-	10% abdominal pain, 3% fatigue/nausea, >20% lymphocytopenia, 5% radiation-induced gastric ulceration, 2% radiation pneumonitis, 2% liver abscess, 2% cholangitis, 50% liver enzyme elevation, <5% liver failure (2–3% fatal)	[48,49,50]
Everolimus	44	not reached	not reached (63% at 24 months)	mainly hematotoxicity (thrombocytopenia, anemia) in the 10 mg/d everolimus dose group 100%, one case (6%) hepatotoxicity	[51]
[^131^I]I-MIBG	0	-	-	one case of three (33%) thrombocytopenia	[52]

ORR = objective response rate; OS = overall survival; PFS = progression-free survival; SAE = serious adverse events according to CTCAE; Cave! In the table are listed collective SAE from different references. These SAE correlate only with the studied cohort in the particular investigation; Ref = references; MDS = myelodysplastic syndrome; CAPTEM = capecitabine and temozolomide; SSA = somatostatin receptor analogues; [^131^I]I-MIBG = [^131^I]I-metaiodobenzylguanidine; EBRT = fractionated external beam radiotherapy.

## Data Availability

Not applicable.

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
