# Peer review of "Combination Therapies with PRRT"

_pharmaceuticals, 2021, doi:10.3390/ph14101005_

Round 1

Reviewer 1 Report

It is a very important publication on combined peptide radionuclide therapy of neuroendocrine tumors. The authors describe the results of work on the combination of radionuclide therapy with non-radiolabeled somatostatin analogues, chemotherapy (capecitabine/temozolomide), molecular targeted treatment (Everolimus), [131I]I-Metaiodobenzylguanidine, external beam radiotherapy and combination of 177Lu and 90Y labeled somatostatine analouges .The paper describes both the biological basics of combination therapy and the results of conducted clinical trials. This work can be very useful for neuroendocrine tumor clinicians to choose the most effective treatment strategy. 

I only have very little comments. The review  should contain information on other effective radionuclide therapies of neuroendocrine tumors such as the use of alpha emitters (225Ac, 213Bi) or Auger emitters (111In). For example combination of alpha and beta radionuclide therapy (Peptide Receptor Radionuclide Therapy Using Ac-225-DOTATOC Achieves Partial Remission in a Patient With Progressive Neuroendocrine Liver Metastases After Repeated beta-Emitter Peptide Receptor Radionuclide Therapy, Clin. Nucl.Med. 45,  241-243, 2020. The second remark concerns references. Authors in many (17) cases do not use journal abbreviations.

Author Response

…This work can be very useful for neuroendocrine tumor clinicians to choose the most effective treatment strategy.

I only have very little comments. 

1) The review should contain information on other effective radionuclide therapies of neuroendocrine tumors such as the use of alpha emitters (225Ac, 213Bi) or Auger emitters (111In). For example combination of alpha and beta radionuclide therapy (Peptide Receptor Radionuclide Therapy Using Ac-225-DOTATOC Achieves Partial Remission in a Patient With Progressive Neuroendocrine Liver Metastases After Repeated beta-Emitter Peptide Receptor Radionuclide Therapy, Clin. Nucl.Med. 45,  241-243, 2020.

Our response: We added this information as suggested (page 6-7, Chapter “Dual PRRT”, red marked text).

2) The second remark concerns references. Authors in many (17) cases do not use journal abbreviations.

Our response: We sent the manuscript for a final check to an editing service and their Quality Assurance Department found that the document met their standards with high marks.

Reviewer 2 Report

The authors reviewed the combination therapies with PRRT. However, some parts do not provide enough information for readers, so improvements are needed.

1) Immuno checkpoint inhibitors are one candidate drug. Would you please check out the following paper? Phase I study of the 177Lu-DOTA0-Tyr3- Octreotate (lutathera) in combination with nivolumab in patients with neuroendocrine tumors of the lung (J Immunother Cancer 2020;8:e000980. doi:10.1136/jitc-2020-000980)

2) Table1 The information in the table is insufficient and difficult to understand. 

-Readers can't know which trial is the Phase1/2 studies. 

-The number of patients is missing. 

-NET type is not described. 

-How do you describe the SAE when there are multiple references? 

I agree with the authors that combination therapies with PRRT should be developed with safety. A detailed description of the side effects would also be helpful to the reader. For example, there was a rare case of a NET patient who develops neutropenic sepsis after PRRT with CAPTEM (A rare case of a patient with a high-grade neuroendocrine tumor developing neutropenic sepsis after receiving PRRT combined with Capecitabine or Temozolomide: A case report, MOLECULAR AND CLINICAL ONCOLOGY 14: 20, 2021). 

3) Figure2

-PRRT has been performed to prolong survival prognosis, as the authors mentioned. However, the prognosis is one factor of a patient in the figure. I got the impression that they were contradictory.

-Please think about whether you should add genetic information to the figure.

4) The authors concluded that combination therapies with PRRT should be investigated further, but is there enough preclinical evidence to support this for all combination therapies? If you have any thoughts on this, please add.

Author Response

The authors reviewed the combination therapies with PRRT. However, some parts do not provide enough information for readers, so improvements are needed.

1) Immuno checkpoint inhibitors are one candidate drug. Would you please check out the following paper? Phase I study of the 177Lu-DOTA0-Tyr3- Octreotate (lutathera) in combination with nivolumab in patients with neuroendocrine tumors of the lung (J Immunother Cancer 2020;8:e000980. doi:10.1136/jitc-2020-000980)

Our response: We have added this information as suggested (page 11, Chapter “Promising Future Combination Therapies”, red marked text).

2) Table1 The information in the table is insufficient and difficult to understand. 

-Readers can't know which trial is the Phase1/2 studies. 

-The number of patients is missing. 

-NET type is not described. 

-How do you describe the SAE when there are multiple references? 

I agree with the authors that combination therapies with PRRT should be developed with safety. A detailed description of the side effects would also be helpful to the reader. For example, there was a rare case of a NET patient who develops neutropenic sepsis after PRRT with CAPTEM (A rare case of a patient with a high-grade neuroendocrine tumor developing neutropenic sepsis after receiving PRRT combined with Capecitabine or Temozolomide: A case report, MOLECULAR AND CLINICAL ONCOLOGY 14: 20, 2021). 

Our response: The aim of Table 1 is to summarize the efficacy and safety data of different combination therapies. We agree that this table does not contain all the information regarding each particular study. More detailed information could be found in the manuscript. Furthermore, we provided a list with the references of the cited papers in the table, so readers could check up data such as study design, methods, inclusion / exclusion criteria and further results such as number of included patients.

In our opinion more details would make the table for most of the clinicians unreadable. To better specify the purpose of the table, we changed the title: “Efficacy and safety of combination treatment with PRRT”.

Regarding SAE we added the following information: “Cave! In the table are listed collective SAE from different references. These SAE correlate only with the studied cohort in the particular investigation.”

We also cited the paper with this interesting case (A rare case of a patient with a high-grade neuroendocrine tumor developing neutropenic sepsis after receiving PRRT combined with Capecitabine or Temozolomide: A case report, MOLECULAR AND CLINICAL ONCOLOGY 14: 20, 2021), as suggested (page 7-8, Chapter “Chemotherapy”, red marked text).

3) Figure2 (now Figure 1)

-PRRT has been performed to prolong survival prognosis, as the authors mentioned. However, the prognosis is one factor of a patient in the figure. I got the impression that they were contradictory.

-Please think about whether you should add genetic information to the figure.

Our response: Thank you for your remark. We specified the term prognosis as life expectancy of at least

3 months.

Furthermore, we added genetic factors of the patient as factors that influence decision-making regarding treatment.

4) The authors concluded that combination therapies with PRRT should be investigated further, but is there enough preclinical evidence to support this for all combination therapies? If you have any thoughts on this, please add.

Our response: Thank you for this comment! We have changed parts of the conclusion as suggested (page 1, Chapter “Conclusion”, red marked text).

(x) English language and style are fine/minor spell check required

Our Response: We sent the manuscript for a final check to an editing service and their Quality Assurance Department found that the document met their standards with high marks.